# Evaluation of Dihydroartemisinin–Piperaquine Efficacy and Molecular Markers in Uncomplicated Falciparum Patients: A Study across Binh Phuoc and Dak Nong, Vietnam

**DOI:** 10.3390/medicina60061013

**Published:** 2024-06-20

**Authors:** Thu Huyen Thi Tran, Bui Thi Thu Hien, Nguyen Thi Lan Dung, Nguyen Thi Huong, Tran Thanh Binh, Nguyen Van Long, Nguyen Dang Ton

**Affiliations:** 1Institute of Biomedicine and Pharmacy, Vietnam Military Medical University, Hanoi 100000, Vietnam; tranthuhuyen.cnsh@gmail.com (T.H.T.T.); bthienhvqy@gmail.com (B.T.T.H.); dung2711997@gmail.com (N.T.L.D.); nguyenvanlong.ckhqs@gmail.com (N.V.L.); 2Department of Biotechnology, Graduate University of Science and Technology, Vietnam Academy of Science and Technology, Hanoi 100000, Vietnam; 3National Burn Hospital, Vietnam Military Medical University, Hanoi 100000, Vietnam; 4103 Hospital, Vietnam Military Medical University, Hanoi 100000, Vietnam; thanhbinh7713@gmail.com; 5Institute of Genome Research, Vietnam Academy of Science and Technology, Hanoi 100000, Vietnam

**Keywords:** artemisinin resistance, ACTs, DHA–PPQ, *P. falciparum*, *pfpm2*, *pfK13*, malaria

## Abstract

*Background and Objectives*: Malaria continues to be a significant global health challenge. The efficacy of artemisinin-based combination therapies (ACTs) has declined in many parts of the Greater Mekong Subregion, including Vietnam, due to the spread of resistant malaria strains. This study was conducted to assess the efficacy of the Dihydroartemisinin (DHA)–Piperaquine (PPQ) regimen in treating uncomplicated *falciparum* malaria and to conduct molecular surveillance of antimalarial drug resistance in Binh Phuoc and Dak Nong provinces. *Materials and Methods*: The study included 63 uncomplicated malaria falciparum patients from therapeutic efficacy studies (TES) treated following the WHO treatment guidelines (2009). Molecular marker analysis was performed on all 63 patients. Methods encompassed Sanger sequencing for *pfK13* mutations and quantitative real-time PCR for the *pfpm2* gene. *Results*: This study found a marked decrease in the efficacy of the DHA-PPQ regimen, with an increased rate of treatment failures at two study sites. Genetic analysis revealed a significant presence of *pfK13* mutations and *pfpm2* amplifications, indicating emerging resistance to artemisinin and its partner drug. *Conclusions*: The effectiveness of the standard DHA-PPQ regimen has sharply declined, with rising treatment failure rates. This decline necessitates a review and possible revision of national malaria treatment guidelines. Importantly, molecular monitoring and clinical efficacy assessments together provide a robust framework for understanding and addressing detection drug resistance in malaria.

## 1. Introduction

Malaria remains a potentially fatal disease caused by parasites transmitted through the bite of *Anopheles* mosquitoes [1]. The World Malaria Report 2023 indicates that nearly half of the world’s population lives in areas at risk of malaria [2]. The Greater Mekong Subregion (GMS), including Vietnam, is one of these critical regions, where high incidences of malaria, predominantly caused by the *Plasmodium falciparum* species, are a continuous challenge [2,3].

Chloroquine was historically the primary drug for treating all forms of malaria, including those caused by *P. falciparum* [4,5,6]. However, *P. falciparum* developed resistance to chloroquine through mutations in the *pfcrt* gene [4,7]. The spread of chloroquine-resistant *P. falciparum* led to significant treatment failures, as well as increased morbidity and mortality rates [4]. Due to these failures, the World Health Organization (WHO) and national health authorities recommended discontinuing chloroquine for treating *falciparum* malaria in favor of more effective alternatives [5,6].

In response, artemisinin-based combination therapies (ACTs) have become a cornerstone of the treatment arsenal [8,9]. Since 2005, the Dihydroartemisinin–Piperaquine (DHA-PPQ) regimen, one of the six ACTs recommended by the World Health Organization (WHO) for treating uncomplicated *P. falciparum* malaria, has been used in Vietnam [9,10]. This regimen combines DHA, a derivative of artemisinin, with PPQ, a bisquinoline [8,11,12]. DHA acts swiftly to reduce the number of malaria parasites in the bloodstream, providing rapid symptom relief [9,13,14]. Piperaquine, with its longer half-life, helps clear residual parasites and provides prolonged protection against recrudescence [9,14].

The *pfk13* (*Plasmodium falciparum Kelch 13*) gene, which encodes the Kelch protein is located on chromosome 13 of *P. falciparum* [15,16,17,18]. The *pfk13* gene comprises three exons and two introns, encoding a protein with multiple Kelch repeats that form a β-propeller structure [18]. This structure is believed to facilitate the protein–protein interactions essential for its role in the cellular stress response [19]. The gene’s expression is highly stage-specific, with the highest levels observed during the ring stage of the parasite’s intraerythrocytic cycle [20]. This stage-specific expression is significant as it correlates with the period when the parasite is most susceptible to artemisinin, highlighting the gene’s role in mediating drug resistance [21].

Mutations in the *pfK13* gene are key indicators of artemisinin resistance, as they can lead to delayed parasite clearance or the presence of parasites three days post-treatment following ACTs [9]. The WHO has reported over 200 mutations in the *pfk13* gene, but only some of these mutations, such as C580Y, Y493H, R539T, I543T, and F446I, are strongly associated with artemisinin resistance [9]. The identification of *pfK13* mutations in a malaria endemic region serves as an early warning sign for the emergence of artemisinin resistance in GMS, including Vietnam [9].

The *pfpm2* (*plasmepsin2*) gene, located on chromosome 14 of *P*. *falciparum*, encodes an aspartic protease involved in the hemoglobin digestion process within the digestive vacuole of the parasite [22,23,24]. An amplification of/copy number variation (CNV) in this gene leads to higher levels of the plasmepsin 2 enzyme being produced [24,25]. Elevated levels of this enzyme can interfere with the effectiveness of antimalarial drugs, specifically those targeting the digestive vacuole such as PPQ [25]. The presence of multiple copies of the *pfpm2* gene correlates strongly with the reduced efficacy of PPQ and thus serves to monitor PPQ resistance [24,26,27,28,29]. A high prevalence of PPQ resistance can also lead to an increased risk of treatment failure for the DHA-PPQ regimen [27].

In 2023, the number of malaria cases in Vietnam was reported to be 448, with Binh Phuoc accounting for 5 cases and Dak Nong for 4 cases [30]. However, two provinces, located in southern Vietnam, are recognized as malaria hotspots due to the complex nature of malaria transmission and the prevalence of resistance in these areas [3,31]. Moreover, a reduced efficacy of the DHA-PPQ regimen has been documented in these provinces [9,32,33]. In this context, monitoring the effectiveness of DHA-PPQ, a commonly used antimalarial treatment regimen, becomes crucial.

This study aims to assess the effectiveness of the DHA-PPQ regimen for treating *P. falciparum* malaria in Binh Phuoc and Dak Nong provinces and to evaluate the prevalence of mutations in the *pfK13* gene and *pfpm2* gene amplification.

## 2. Materials and Methods

### 2.1. Study Sites and Patients

A total of 63 uncomplicated falciparum malaria samples were collected from Dak Nong (n = 19, August 2018 to May 2019) and Binh Phuoc (*n* = 44, August 2018 to May 2019). Patients included in the study had a mono-infection with *P. falciparum* malaria. The parasitemia ranged from 1000 to 200,000 per µL. The inclusion criteria also required a fever (defined as a temperature ≥ 37.5 °C) or a history of fever. Eligible participants were males or non-pregnant females, aged between 2 years and 70 years. Written informed consent was obtained from patients, or from parents/guardians in the case of minors; those unable to write provided consent in the presence of a witness.

Patients with symptoms of severe or complicated malaria [34], pregnancy or lactation, infection with other malaria species, severe malnutrition, and with known hypersensitivity or contraindications to the artemisinins or to piperaquine were excluded.

### 2.2. Treatment and Follow-Up

Eligible patients were treated with a full three-day course of DHA-PPQ and followed up for 42 days according to a Therapeutic Effective Study (TES) protocol recommended by the WHO [35]. Each DHA–PPQ tablet contains 40 mg of DHA and 320 mg of PPQ and is commercially known as Arterakin^®^, produced by Central Pharmaceutical Factory No. 1 (Hanoi, Vietnam), batch number 17001. The daily drug dosage was determined based on the patient’s age group; the correct drug dosage was ascertained using the Table 1.

All medication doses were given under the direct supervision of the study team. Patients were observed for 30 min to 1 h after each dose to monitor for vomiting or other side effects. If vomiting occurred within 30 min after administration, the full dose was re-administered. If vomiting occurred between 30 min and 1 h after administration, half of the dose was re-administered. Patients who vomited a second time were withdrawn from the study and received parenteral therapy according to national guidelines. A standard dose of primaquine (0.25 mg/kg daily for 2 weeks) was given to patients at the end of the follow-up period or at recurrence.

All enrolled patients were followed for 42 days, with scheduled visits on days 1, 2, 3, 7, 14, 21, 28, 35, and 42. Additionally, patients were instructed to return on any day they experienced symptoms of illness to allow for the monitoring of clinical recovery or recurrence of malaria symptoms. The day of enrollment and the first dose of medication is referred to as day 0. During follow-up visits, clinical assessments were conducted, axillary temperatures were measured, and blood slides were taken for parasite counts. On day 0 and on the day symptoms recurred, dried blood spots were collected for genotyping to distinguish between reinfection and recrudescence. Recrudescence is defined as the patient being reinfected with the same strain of parasite from the initial infection, while reinfection occurs when a patient contracts a different strain of the malaria parasite.

Treatment outcomes were classified on day 42 after DHA-PPQ regimen following the WHO protocol as follows: (1) early treatment failure (ETF) if the patient presented with parasitemia and signs of danger on day 1, 2 or 3 or persistent parasitemia until day 3; (2) late clinical failure (LCF) was defined as the presence of signs of danger or severe malaria, or an axillary temperature of  ≥37.5 °C with parasitemia between days 4 and 42 in a patient who did not qualify as early treatment failure; (3) late parasitological failure (LPF) if a patient had parasitemia between days 7 and 42 in the absence of fever or other clinical symptoms, and was not classified as early treatment failure; (4) adequate clinical and parasitological response (ACPR) in the absence of parasitemia in a patient who was not classified as early, late clinical, or late parasitological failure; (5) lost to follow-up when, despite all reasonable efforts, an enrolled patient did not attend the scheduled visits and could not be found, and thus the patient was withdrawn from the study; and (6) withdrawal when the patient consented to withdraw, failed to complete treatment or was reinfected.

### 2.3. Ethics

The studies received ethical approval from the Vietnam Military Medical University Committee on Morality in Medical Biological Research (Approval No. 1690/GCN-HVQY). Written informed consent was obtained from all participants, as well as from the parents or guardians of minor participants, prior to enrollment.

### 2.4. Genomic DNA Extraction

Dried blood spots were collected and genomic DNA was extracted using the GeneJET Whole Blood Genomic DNA Purification Kit (Thermo Fisher, Waltham, MA, USA) according to the manufacturer’s instructions. The concentration and quality of the extracted DNA were assessed using a NanoDrop 2000 spectrophotometer (Thermo Fisher, Waltham, MA, USA). DNA purity was evaluated by measuring the A260/A280 ratio, with a range of 1.7–2.0 generally considered pure for DNA. The extracted DNA was stored at –20 °C until being amplified by PCR.

### 2.5. Genotyping of Malaria Parasites to Differentiate between Recrudescence and Reinfection

The *msp1*, *msp2*, and *glurp* genes were amplified using a nested PCR strategy to differentiate between recrudescence and reinfection in accordance with the WHO recommendations [36]. The primer pairs used were as previously described [37]. The *P. falciparum* 3D7 strain, provided by the National Institute of Malariology Parasitology and Entomology (Hanoi, Vietnam), was used as the positive control, while nuclease-free water served as the negative control. Briefly, genomic DNA served as the template in a PCR reaction. The reaction mixture for the first round of PCR included 3 μL of extracted DNA, 25 μL of dNTPs at 20 mM, 25 μL of 10× PCR buffer containing 20 mM MgCl_2_, 10 μL each of forward and reverse primers at 125 μM, and 10 μL of Taq-DNA polymerase at 1 unit per μL. The thermal cycling conditions for the first round were as follows: initial denaturation at 95 °C for 5 min; followed by 25 cycles of 94 °C for 1 min, 58 °C for 2 min, and 72 °C for 2 min; and a final extension at 72 °C for 8 min. For the second round of PCR, 1 μL of the diluted first-round product was used as a template. The reaction mixture was similar to that of the first round, but included inner primers specific to the subtypes of each gene. The thermal cycling parameters for the second round included the following: initial denaturation at 95 °C for 5 min; followed by 30 cycles of 94 °C for 1 min, 61 °C for *msp2* gene or 58 °C for *msp1* gene and *glurp* gene for 2 min, and 72 °C for 2 min; and ending with a final extension at 72 °C for 8 min. The amplified PCR products were then subjected to agarose gel electrophoresis using a 2% gel stained with ethidium bromide to visualize the DNA bands under UV light. Electrophoresis was performed using 0.5× TBE buffer to ensure the optimal separation and resolution of the DNA fragments. Specific band sizes corresponding to each gene’s genotypes were identified, facilitating the determination of genetic profiles indicative of either recrudescence or reinfection.

### 2.6. Detection of pfK13 Gene Mutations

The *pfK13* gene was amplified using a nested PCR approach, as described previously [38]. Briefly, two μL of genomic DNA were used as a template to amplify target genes. The amplification process involved 10 μL of DreamTaq Master Mix (2×) and 0.5 μL of each primer at a concentration of 10 pmol/μL for both the primary and secondary reactions. For the primary reaction, the following cycling parameters were used: 5 min at 94 °C, 25 cycles at 94 °C for 30 s, 51 °C for 60 s, 72 °C for 1 min 25 s, and a final extension for 5 min at 72 °C. For the nested PCR, 1 μL of 1/5 diluted primary PCR product was used as a template. Thermal cycling for the secondary PCR reaction included 5 min at 95 °C, 30 cycles at 94 °C for 45 s, 50 °C for 45 s, 72 °C for 45 s, and a final extension for 5 min at 72 °C. The PCR products were visualized after electrophoresis on a 2% agarose gel stained with ethidium bromide. Electrophoresis was performed using 0.5× TBE buffer to ensure the optimal separation and resolution of the DNA fragments. Secondary PCR products were purified by the GeneJET PCR Purification Kit (Thermo Scientific, Waltham, MA, USA) and sequenced (Macrogen Inc., Seoul, Republic of Korea). The sequences were aligned with those of 3D7 retrieved from the *P. falciparum* database and deposited in GenBank (Accession No. NC_004331.3).

### 2.7. Quantitative PCR to Assess pfpm2 Gene Copy Number

Gene copy number variation was determined using real-time PCR (Rotor-GeneQ, Qiagen, Hilden, Germany) using a previously reported protocol [39]. The reaction was carried out in 25 μL containing 12.5 μL of 2× QuantiTect Probe RT-PCR Master Mix (Qiagen, Hilden, Germany), 0.4 µM of each forward reverse pfpm2 primer, 0.1 µM of pfpm2 probe, 0.4 µM of each forward and reverse β-tubulin primer and 0.1 µM β-tubulin probe, and 2 μL of template DNA and water. The cycling parameters were 95 °C for 15 min, followed by 45 cycles at 94 °C for 15 s, and finally 94 °C for 15 s using a Rotor GeneQ (Qiagen, Germany). All samples were run in duplicate. The 3D7 clones were used as single-copy calibrators. Copy numbers were calculated using the following formula: copy number  =  2^−ΔΔCt^, with ΔΔCt denoting the difference between the ΔCt of the unknown sample and the ΔCt of the reference sample (3D7) [40]. Multiple copies vs. a single copy of *pfpm2* were defined as copy numbers < 1.5 and ≥1.5, respectively.

### 2.8. Statistical Analysis

Data management was conducted using Excel 2016, with descriptive statistics such as percentages, means, medians, standard deviations, and ranges reported as appropriate. Continuous variables, including parasitemia, age, weight, and height, were compared between the two sites using a *t*-test for normally distributed data or a Wilcoxon rank sum test for non-normally distributed data. The normality of the data was assessed using the Shapiro–Wilk test. If the data were normally distributed, a t-test was used for comparisons; if not, the Wilcoxon rank sum test was applied. Discrete variables, such as gender, were compared using a Chi-square test. The rate of patients still exhibiting parasitemia after three days of treatment was calculated. Treatment outcomes were defined according to the WHO guidelines, with the proportions of ACPR, ETF, LCF, and LPF by day 42 presented descriptively for both study sites. Kaplan–Meier survival curves were utilized to estimate the probability of recrudescence within 42 days. The frequency of mutations was determined through simple counting methods and mutation frequencies were calculated and presented as frequencies and percentages. All analyses were performed using R v.3.3.1 software (R Foundation for Statistical Computing, Vienna, Austria).

## 3. Results

### 3.1. Baseline Characteristics

The general characteristics of the study population are presented in Table 2. In Binh Phuoc, the mean age was 30.5 ± 9.1 and the sex ratio was 42 males to 2 females. All exhibited fever (axillary temperature ≥ 37.5 °C). The mean parasitemia stood at 16,659/µL (560 to 95,428/µL). In Dak Nong, the population was predominantly male (94.7%, 18/19. The mean age of the patients was 31.5 ± 9.8, the mean axillary temperature was 38.3 ± 0.6, and parasitemia averaged at 21,871/µL (1000 to 154,666/µL).

### 3.2. Therapeutic Efficacy of DHA-PPQ for Treatment of Uncomplicated P. falciparum

Table 3 presents data on the temperature and parasitemia levels of patients at two study sites, Binh Phuoc and Dak Nong, over the first 3 days of treatment (day 1 to day 3). In both sentinel sites, the incidence of fever among patients showed a marked decline from day 1 to day 3. By day 3, the incidence of fever had completely resolved at both locations, with no cases reported.

In Binh Phuoc and Dak Nong, the rate of parasitemia significantly decreased from day 1 to day 3. However, in Binh Phuoc, 22.7% (10 out of 44) of patients remained parasitemic by day 3, and in Dak Nong, 18.8% (3 out of 16) of patients still exhibited parasitemia.

According to the WHO guidelines, suspected endemic artemisinin resistance is defined as persistent parasitemia in ≥10% of patients, as detected by microscopy on day 3 after treatment with ACTs or artesunate monotherapy. Although there was a significant decrease in parasitemia from day 1 to day 3 in both Binh Phuoc and Dak Nong, the day 3 positivity rates still exceeded this threshold, with 22.7% in Binh Phuoc and 18.8% in Dak Nong, suggesting potential artemisinin resistance.

Additionally, we observed a significant number of patients experiencing the recurrence of parasites, predominantly between day 7 and day 42 post-treatment, with 12 cases reported in Binh Phuoc and 3 cases in Dak Nong. However, a genotyping approach confirmed that three of these cases were reinfections. Patients infected with the new strain were withdrawn from the study. The results are presented in Table 4.

The Kaplan–Meier survival curve, used to estimate the probability of recrudescence within 42 days, reveals a notably high cumulative recrudescence rate in the regions studied. Specifically, the data indicate that the recrudescence rate in Binh Phuoc reached 23.7% (9 out of 38), while in Dak Nong, it was slightly lower at 20% (3 out of 15). High recrudescence rates indeed serve as a critical indicator of treatment failure in malaria management (Figure 1).

By day 42 after the DHA-PPQ regimen, treatment classification was conducted following the WHO guidelines [35]. The treatment outcomes at the two study sites are presented in Table 5. Of the 63 patients enrolled, 53 completed the 42-day follow-up after treatment with DHA-PPQ. Three patients were withdrawn due to reinfection, while seven were lost to follow-up. No ETFs were recorded in either region.

The treatment failure rates revealed significant findings in both locations: Binh Phuoc recorded a high late LCF rate of 31.8%, compared to only 5.3% in Dak Nong. In contrast, LPF rates were observed at 6.8% in Binh Phuoc and were higher in Dak Nong at 15.8%. Furthermore, statistical analysis revealed a significant difference in treatment outcomes between Binh Phuoc and Dak Nong (*p* = 0.042), particularly in terms of treatment failure rates.

Both sentinel sites reported relatively low cure rates, with ACPR rates of 47.7% in Binh Phuoc and 57.9% in Dak Nong, resulting in an average rate of 50.8%. These data indicate a decrease in the treatment efficacy of the DHA-PPQ regimen at both study sites.

### 3.3. Molecular Marker Surveillance

Table 6 shows the prevalence of the *pfK13* gene mutation and the copy number variation in the *pfpm2* gene at the two study sites.

We genotyped the *pfk13* gene by performing Sanger sequencing on 63 samples collected between August 2018 and May 2019 in Binh Phuoc and Dak Nong provinces. However, only 61 of these samples were successfully sequenced. A total of 49 samples bearing the *pfk13* C580Y mutation were identified, representing a prevalence of 80.3% (49 out of 61).

Furthermore, the prevalence of the C580Y mutation varied between the study locations, with Dak Nong presenting a prevalence of 67% and Binh Phuoc showing a higher prevalence of 86%. It is remarkable that, despite the higher prevalence of the C580Y mutation in Binh Phuoc, statistical analysis indicated that the observed differences in mutation rates between the two study sites were not statistically significant (*p* > 0.05).

We used real-time quantitative PCR to validate the copy number variation in the *pfpm2* gene in 63 collected samples. However, only 51 out of 63 samples yielded conclusive results. The presence of multiple *pfpm2* copies was confirmed in 24 infections, accounting for 47% (24 out of 51) of the successfully analyzed samples. In detail, multiple copies were found in 54% (7 out of 13) of infections in Dak Nong and 50% (19 out of 38) in Binh Phuoc. There was no statistically significant difference in the proportion of isolates with multiple copies of *pfpm2* between the two study sites (*p* > 0.05).

Our study also observed a high prevalence of parasites carrying both the C580Y mutation and multiple copies of the *pfpm2* gene. Specifically, 35.1% of cases in Binh Phuoc and 38.5% in Dak Nong exhibited this combined genotype.

## 4. Discussion

Regular monitoring of the therapeutic efficacy of ACTs is crucial for making timely adjustments to treatment policies and detecting early shifts in *P. falciparum* susceptibility to antimalarial medications [9,14,35]. The WHO recommends that all falciparum-endemic countries monitor the efficacy of first-line and second-line ACTs every two years [35]. This therapeutic efficacy study provided critical data, including the percentage of patients with parasitemia on day 3, the current preferred indicator for routinely monitoring suspected artemisinin resistance in *P. falciparum*, and the percentage of treatment failures at follow-up on either day 28 or day 42, depending on the half-life of the partner drug in the ACT [9]. If the treatment failure rate reaches or exceeds 10%, it is crucial to revise the national antimalarial treatment policy [9,35].

DHA-PPQ, a cornerstone antimalarial therapy, has played a critical role in managing uncomplicated *P. falciparum* malaria in Vietnam since 2005 [11]. Originally, DHA-PPQ showcased outstanding success rates, exemplified by the 2007 reports from Binh Phuoc province in Vietnam, where the ACPR was an impressive 100%, and the day 3 positivity rate was a mere 4% [41]. However, this efficacy began to wane significantly in the following years. By 2009, the ACPR had declined slightly, and the day 3 positivity rates had escalated to 15.3% [41]. This downward trend continued sharply, with the day 3 positivity rate reaching 30.6% by 2012 and 36% by 2013 [42]. By 2015, the effectiveness of the therapy had markedly deteriorated, culminating in an ACPR of just 68.2% [43]. Similar patterns of declining efficacy and delayed parasite clearance were also observed in other malaria-endemic regions within Vietnam, such as Gia Lai, Dak Nong, Quang Nam, Khanh Hoa, and Ninh Thuan provinces [9]. Notably, Gia Lai province experienced a steady decrease in ACPR from 2012 to 2016, coupled with increasing day 3 positivity rates, indicating a significant loss in the effectiveness of DHA-PPQ [32].

Our study evaluating the efficacy of DHA-PPQ in two provinces in southern Vietnam revealed particularly alarming trends that align with previous research [32,41,44]. Specifically, the ACPR rates plummeted to 47.7% in Binh Phuoc province and 57.9% in Dak Nong province. Concurrently, high day 3 positivity rates persisted, with 22.7% in Binh Phuoc province and 18.8% in Dak Nong province. These findings highlight a critical and substantial decline in the therapeutic efficacy of DHA-PPQ, evidenced by the sharp drop in cure rates and significant increases in day 3 positivity rates. This ACT should be discontinued immediately for treating falciparum malaria. The continued use of DHA-PPQ is likely to increase selective pressure on *P. falciparum* parasites in these regions, potentially leading to the dominance of resistant parasites throughout the country. Alternative treatments should be evaluated without delay.

A trend toward the reduced efficacy of the DHA-PPQ regimen has also been observed in neighboring countries in the eastern GMS region, including Cambodia [45], Laos [46], and Thailand [17,46]. This decline is mainly attributed to the emergence and spread of drug-resistant strains of *P. falciparum* [47], marked by specific genetic mutations such as those in the *pfK13* [16,47,48,49] gene and amplifications of *pfpm2* [33,46,47,50].

Shifting focus to the genetic landscape, our results show that C580Y was the only mutation on the *pfK13* gene found, with a high prevalence of up to 80.3% of parasites carrying this validating artemisinin-resistant mutation. To date, 13 distinct non-synonymous mutations of the *pfK13* gene have been identified in Vietnam [33]. Among these, the C580Y and I543T mutations were the most common. Notably, by 2015, the C580Y mutation became predominant across all malaria-endemic regions in Vietnam, specifically replacing the initially prevalent I543T mutation in the Central Highlands [33]. Moreover, there has been a significant increase in the frequency of validated *pfK13* mutations over time, rising from 28.6% in the early 2000s to 69% by 2015–2016 [33]. Similarly, the amplification of the *pfpm2* gene has steadily increased over time, rising from 3.2% in the early 2000s to 10.9% by the period 2015–2016 [33], and reaching up to 47% in our study.

The C580Y mutation in the *pfk13* gene has also emerged as predominant in other GMS countries [50]. The prevalence of *pfK13* mutations is dynamic, illustrating the evolving nature of malaria and the parasite’s adaptation to antimalarial drugs. The limited variety in detected mutations suggests a selective pressure that favors the C580Y mutation, likely contributing to its dominance in the parasite population [50].

Notably, high proportions of *pfpm2* gene amplification have frequently been observed in several African sites, particularly in Burkina Faso and Uganda, where rates exceeded 30% [28]. However, the appearance of *pfpm2* gene amplification across diverse and distant locations in Africa suggests possible independent emergence rather than a spread from GMS [28].

Additionally, a significant proportion of cases (36%) exhibited both the C580Y mutation and multiple *pfpm2* copies, hinting that multidrug resistance might be emerging in Vietnam. The emergence of multidrug resistance in Vietnam, where DHA-PPQ is heavily used, explains the rapid decline of this regimen in the region [33,50,51]. The emergence of multidrug resistance in Vietnam, particularly concerning malaria, is intricately linked to the patterns of resistance development observed in Cambodia [50,51], where resistance to artemisinin and its partner drugs suggests a two-stage selection process [47,51]. Initially, the spread of artemisinin resistance led to reduced parasite genetic diversity, with the majority of parasites carrying *pfK13* C580Y [17,52]. This was followed by the emergence of PPQ resistance, which developed from those genetic backgrounds [17,51,52]. Subsequently, parasite lineages with the *pfK13* C580Y mutation and multiple *pfpm2* copies (or *PfPailin* lineage) spread to the southeastern parts of Thailand [23,49] and Champasak in Laos [17,47]. Later, they also moved rapidly to the southern part of Vietnam [44,51], demonstrating a strong selective sweep within a brief period [17,47].

The observed decrease in therapeutic effectiveness aligns with the detection of significant genetic markers of resistance to both components of this combination therapy. The correlation between clinical outcomes of treatment and resistance markers highlights the critical role of genetic surveillance in guiding malaria treatment strategies. Therefore, the molecular monitoring of drug resistance mutations plays a critical role not only in providing early warnings about the emergence of resistance but also in tracking the spread of resistant strains across geographical regions. This surveillance helps identify the specific genetic mutations associated with resistance to antimalarial drugs, enabling healthcare professionals and researchers to observe the development and proliferation of resistant mutations. Moreover, a geographic analysis of resistance patterns is crucial, especially in border areas or regions with high mobility, as it can reveal the pathways through which resistance spreads between populations.

## 5. Conclusions

Our findings support the necessity of revising national malaria treatment guidelines and enhancing molecular monitoring to mitigate the spread of resistant strains. The integration of genetic surveillance with clinical outcomes provides a robust framework for understanding and addressing drug resistance in malaria, which is crucial for informing public health strategies and ensuring the effectiveness of malaria control and elimination efforts in the region.

## Figures and Tables

**Figure 1 medicina-60-01013-f001:**
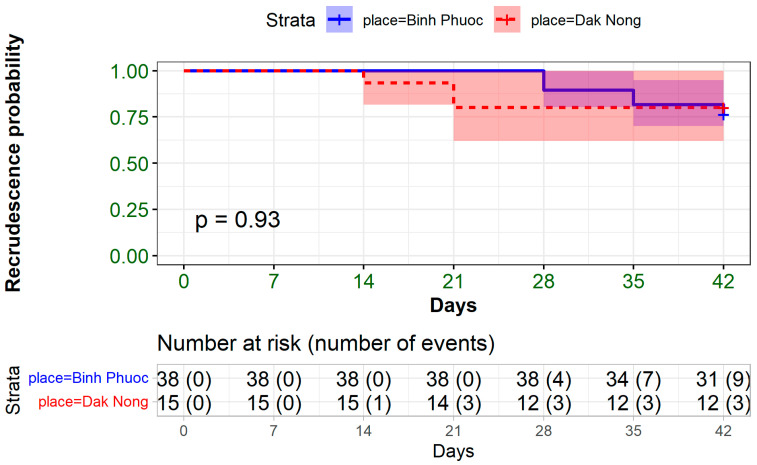
Kaplan–Meier recrudescence probability analysis for Binh Phuoc and Dak Nong in 42-day follow-up.

**Table 1 medicina-60-01013-t001:** Number of DHA-PPQ tablets administered to patients according to age group.

Age Group	Number of DHA-PPQ (Arterakin^®^) Tablets Per Day					
	0 h	8 h	24 h	32 h	48 h	56 h
2–3 years	0.5	0.5	0.5	0.5	0.5	0.5
3–7 years	0.75	0.75	0.75	0.75	0.75	0.75
8–14 years	1.25	1.25	1.25	1.25	1.25	1.25
>15 years	1.5	1.5	1.5	1.5	1.5	1.5

**Table 2 medicina-60-01013-t002:** General characteristics of the study population (*n* = 63).

Characteristic	Binh Phuoc	Dak Nong	*p*-Value
	(*n* = 44)	(*n* = 19)	
Age (years)			0.804 *
$\bar{X}$ ± *SD*	30.5 ± 9.1	31.5 ± 9.8	
Gender			1.0 **
Male	42 (95.5%)	18 (94.7%)	
Female	2 (4.5%)	1 (5.3%)	
Height (cm)			0.318 *
$\bar{X}$ ± *SD*	167 ± 5.6	165.8 ± 5.5	
Weight (kg)	60.7 ± 7.3	56.5 ± 6.1	0.066 *
$\bar{X}$ ± *SD*			
Temperature at day 0 (°C)	38.7 ± 0.6	38.3 ± 0.5	0.011 *
$\bar{X}$ ± *SD*			
Parasitaemia at day 0 (range)	16,659	21,871	0.498 *
	(560–95,428)	(1000–154,666)	

*: Wilcoxon rank sum test; **: Chi-square test.

**Table 3 medicina-60-01013-t003:** Temperature and parasitemia of all patients from day 1 to day 3 after treatment.

Parameters		Binh Phuoc (*n* = 44)	Dak Nong (*n* = 16) *	*p*-Value **
Day	Parameters			
1	Fever	19 (43.2%)	10 (52.6%)	0.678
	Parasitemia	43 (97.7%)	13 (81.2%)	0.054
2	Fever	2 (4.6%)	1 (5.3%)	1.00
	Parasitemia	25 (58.1%)	8 (50.0%)	0.791
3	Fever	0 (0%)	0 (0%)	1.00
	Parasitemia	10 (22.7)	3 (18.8%)	1.0

*: 3 cases lost to follow-up, ** Chi-squared test.

**Table 4 medicina-60-01013-t004:** Genotyping results of the parasites at day 0 and day of recurrence.

No.	Patient ID	Study Sites	Day of Recurrence	Recrudescence/Reinfection
1	D1.40	Dak Nong	21	Recrudescence
2	D1.42	Dak Nong	21	Recrudescence
3	D1.44	Dak Nong	14	Recrudescence
4	D1.45	Binh Phuoc	42	Recrudescence
5	D1.46	Binh Phuoc	35	Recrudescence
6	D1.47	Binh Phuoc	35	Recrudescence
7	D1.49	Binh Phuoc	28	Recrudescence
8	D1.53	Binh Phuoc	35	Recrudescence
9	D1.54	Binh Phuoc	28	Reinfection
10	D1.56	Binh Phuoc	28	Recrudescence
11	D1.59	Binh Phuoc	42	Recrudescence
12	D1.60	Binh Phuoc	21	Reinfection
13	D1.61	Binh Phuoc	28	Recrudescence
14	D1.63	Binh Phuoc	28	Reinfection
15	D1.64	Binh Phuoc	28	Recrudescence

**Table 5 medicina-60-01013-t005:** Treatment outcome from Binh Phuoc and Dak Nong over the 42-day follow-up.

Classification of Treatment		Binh Phuoc	Dak Nong	Overall	*p*-Value *
		*n*, %	*n*, %	*n*, %	
Total patient treatment failure and complete 42-day follow-up	ETF	0 (0)	0 (0)	0 (0)	0.042
	LCF	14 (31.8)	1 (5.3)	15 (23.8)	
	LPF	3 (6.8)	3 (15.8)	6 (9.5)	
	ACPR	21 (47.7)	11 (57.9)	32 (50.8)	
Patient LFU/WTH	LFU	3 (6.8)	4 (21.1)	7 (11.1)	
	WTH	3 (6.8)	0 (0)	3 (4.8)	

ETF: early treatment failure; LCF: late clinical failure; LPF: late parasitological failure; LFU: lost to follow-up; WTH: withdrawn; ACPR: adequate clinical and parasitological response; * Chi-squared test.

**Table 6 medicina-60-01013-t006:** Prevalence of mutation across study sites.

Mutation/CNV	Provinces *n* (%)		Total	*p*-Value
	Binh Phuoc	Dak Nong		
C580Y	37/43 (86.0)	12/18 (66.7)	49/61 (80.3)	0.2
Multiples of *pfpm2* copy numbers	17/38 (44.7)	7/13 (53.8)	24/51 (47.1)	0.6
C580Y/pfpm2	13/37 (35.1)	5/13 (38.5)	18/50 (36.0)	0.9

## Data Availability

Data are contained within the article.

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
