# Peer review of "Evaluation of Dihydroartemisinin–Piperaquine Efficacy and Molecular Markers in Uncomplicated Falciparum Patients: A Study across Binh Phuoc and Dak Nong, Vietnam"

_medicina, 2024, doi:10.3390/medicina60061013_

Round 1
Reviewer 1 Report
Comments and Suggestions for Authors
The comments and suggestions are described in the attached Word document!

Author Response
Comment 1: Introduction, lines 35-39: In the Introduction, the following information needs to be included: For the species Plasmodium falciparum, Chloroquine is not recommended because this parasite has become resistant, and malaria infections caused by this species no longer respond to this drug. For the species Plasmodium falciparum, it is recommended to use 20 mg of artemether and 120 mg of lumefantrine.
Response 1: Thank you for your valuable comments and for highlighting this important information. We have added the recommended details about Chloroquine resistance to the Introduction section. These changes can be found on page 2, lines 42-48. Your suggestions have helped us improve the accuracy and comprehensiveness of our manuscript.
“Chloroquine was historically the primary drug for treating all forms of malaria, including those caused by P. falciparum [4-6]. However, P. falciparum developed resistance to chloroquine through mutations in the pfcrt gene [4,7]. The spread of chloroquine-resistant P. falciparum led to significant treatment failures, as well as increased morbidity and mortality rates [4]. Due to these failures, the World Health Organization (WHO) and national health authorities recommended discontinuing chloroquine for treating falciparum malaria in favor of more effective alternatives [5,6].”
Comment 2: Introduction, line 39: Remember that when citing the scientific name (e.g., Plasmodium falciparum) of the species for the first time, indicate who described it, according to the rules of zoological nomenclature; remember that it should be in italics.
Response 2: Thank you for your valuable comment. We have italicized the scientific name Plasmodium falciparum and indicated who described it, according to the rules of zoological nomenclature. This correction is located on page 1, line 39. Your suggestion has improved the accuracy and adherence to proper nomenclature in our manuscript.
“The Greater Mekong Subregion (GMS), including Vietnam, is one of these critical regions, where high incidences of malaria, predominantly caused by the Plasmodium falciparum species, are a continuous challenge [2,3]”.
Comment 3: Introduction, lines 49-56: In addition to the Gene Name and Identification, and the Genomic Location, please include: *Gene Function: Description of the gene's biological functions. *Gene Structure: Information about the gene's structure, such as the number of exons and introns and relevant structural characteristics. *Gene Expression Details: Details on the gene's expression in different developmental stages, or physiological and pathological conditions. Known Gene Variants: Information on known gene variants, including mutations, polymorphisms.
Response 3: We thank you again for your detailed recommendations. We have expanded the gene description section to include Gene Function, Gene Structure, Gene Expression Details, and Known Gene Variants. These additions can be found on page 2, lines 57-65. Your comprehensive suggestions have significantly enhanced the completeness and accuracy of our manuscript.
“The pfk13 (Plasmodium falciparum Kelch 13) gene, which encodes the Kelch protein is located on chromosome 13 of P. falciparum [15-18]. The pfk13 gene comprises three exons and two introns, encoding a protein with multiple Kelch repeats that form a β-propeller structure [18]. This structure is believed to facilitate protein-protein interactions essential for its role in the cellular stress response [19]. The gene's expression is highly stage-specific, with the highest levels observed during the ring stage of the parasite intraerythrocytic cycle [20]. This stage-specific expression is significant as it correlates with the period when the parasite is most susceptible to artemisinin, highlighting the gene's role in mediating drug resistance [21]”.
Comment 4: Table 1: The table is misaligned with the line numbers! Please fix it.
Response 4: According to your comment, Table 1 was aligned, and the correction can be found on page 3.
Table 1: Number of DHA-PPQ tablets administered to patients according to age group
|
Age group |
Number of DHA-PPQ (Arterakin®) tablets per day |
|||||
|
0 hour |
8 hour |
24 hour |
32 hour |
48 hour |
56 hour |
|
|
2-3 years |
0.5 |
0.5 |
0.5 |
0.5 |
0.5 |
0.5 |
|
3 -7 years |
0.75 |
0.75 |
0.75 |
0.75 |
0.75 |
0.75 |
|
8 – 14 years |
1.25 |
1.25 |
1.25 |
1.25 |
1.25 |
1.25 |
|
>15 years |
1.5 |
1.5 |
1.5 |
1.5 |
1.5 |
1.5 |
Comment 5: Materials and Methods, 2.4. Genomic DNA extraction: Please specify how the quality control of the extracted DNA was performed (including the type of spectrophotometer used for this), the quality, and the DNA quality ratio 260/280.
Response 5: We have complemented the information pertaining to the quality control measures for genomic DNA extraction, including the type of spectrophotometer and the DNA quality ratio 260/280. This information can be found on page 4, section 2.4, line 152-155.
“The concentration and quality of the extracted DNA were assessed using a NanoDrop 2000 spectrophotometer (Thermo Fisher, USA). DNA purity was evaluated by measuring the A260/A280 ratio, with a range of 1.7-2.0 generally considered pure for DNA”.
Comment 6: Materials and Methods, 2.5. Genotyping of malaria parasites to differentiate recrudescence and reinfection: Please specify the positive and negative controls included in each round of PCR to ensure the accuracy of the results and as a quality control measure.
Response 6: We have included details about the positive and negative controls used in each round of PCR in the genotyping section. These details are on page 4, section 2.5, line 160-162.
“The P. falciparum 3D7 strain, provided by the National Institute of Malariology Parasitology and Entomology (Hanoi, Vietnam), was used as the positive control, while nuclease-free water served as the negative control”.
Comment 7: Materials and Methods, 2.8. Statistical analysis lines 120-134: It was reported that continuous variables, including parasitemia, age, weight, and height, were compared between the two sites using a t-test for normally distributed data or the Wilcoxon rank sum test for non-normally distributed data. Include the dilution of the antibodies. Specify which test was used to check the normality of the data. Remember that you can only use parametric tests if your data is normally distributed.
Response 7: Thank you for your detailed comments. We have included the method used to check the normality of the data. These changes can be found on page 5, section 2.8, lines 211-213." Your suggestion has improved the clarity and thoroughness of our statistical analysis section.
"The normality of the data was assessed using the Shapiro-Wilk test. If the data were normally distributed, a t-test was used for comparisons; if not, the Wilcoxon rank sum test was applied”.
Comment 8: Results, Table 1: Remove the parentheses around the standard deviation values and add “±”.
Response 8: Thank you for pointing out this formatting issue. We have removed the parentheses around the standard deviation values and added “±”. The revised table is on page 6. Your attention to detail has helped us improve the presentation of our data.
Table 2: General characteristics of the study population (n = 63)
|
Characteristic
|
Binh Phuoc (N=44) |
Dak Nong (N=19) |
p-value |
|
Age (years) ± SD |
30.5 ± 9.1 |
31.5 ± 9.8 |
0.804* |
|
Gender Male Female |
42 (95.5%) 2 (4.5%) |
18 (94.7%) 1 (5.3%) |
1.0** |
|
Height (cm) ± SD |
167 ± 5.6 |
165.8 ± 5.5 |
0.318* |
|
Weight (kg) ± SD |
60.7 ± 7.3 |
56.5 ± 6.1 |
0.066*
|
|
Temperature at Day 0 (0C) ± SD |
38.7 ± 0.6 |
38.3 ± 0.5 |
0.011* |
|
Parasitaemia at day 0 (range)
|
16 659 (560 – 95 428) |
21 871 (1000-154 666) |
0.498* |
*: Wilcoxon rank sum test; **: Chi-square test
Comment 9: Results, Table 3: Remove the parentheses around the percentages and place the “%” symbol next to the value.
Response 9: Thank you again for pointing out the formatting issue. We have removed the parentheses around the percentages and placed the “%” symbol next to the value. The revised table is on page 6.
Table 3: Temperature and parasitemia of all patients from day 1 to day 3 after treatment
|
Parameters |
Binh Phuoc (N=44) |
Dak Nong (N=16)* |
p-value** |
|
|
Day |
Parameters |
|||
|
1 |
Fever |
19 (43.2%) |
10 (52.6%) |
0.678 |
|
Parasitemia |
43 (97.7%) |
13 (81.2%) |
0.054 |
|
|
2 |
Fever |
2 (4.6%) |
1 (5.3%) |
1.00 |
|
Parasitemia |
25 (58.1%) |
8 (50.0%) |
0.791 |
|
|
3 |
Fever |
0 (0%) |
0 (0%) |
1.00 |
|
Parasitemia |
10 (22.7) |
3 (18.8%) |
1.0 |
|
*: 3 case loss to follow-up, ** Chi-squared test
Comment 10: Results, Table 4: For the Day of recurrence value, please add the “°” symbol as it is an ordinal value.
Response 10: Thank you for your suggestion. To clarify, the "Day of recurrence" values are indeed ordinal, indicating the sequence of recurrence days. However, using the “°” symbol to denote this is not a common practice in our reporting format. Instead, we have clearly labeled the column as "Day of recurrence" to indicate the ordinal nature of these values. We hope this approach maintains clarity and aligns with the standard reporting format.
Reviewer 2 Report
Comments and Suggestions for Authors
Malaria is an important disease with severe consequences for infected individuals, the public health system and countries at risk of this disease. Hence, coupled with the occurrence of drug resistance of Plasmodium parasites, this study “Evaluation of DHA-PPQ Efficacy and Molecular Markers in Uncomplicated Falciparum Patients: A Study across Binh Phuoc and Dak Nong, Vietnam” by Tran and others is very important.
General Comments
The study is well written although there are a few grammatical, typographical and punctuation errors and these have been indicated in the reviewed manuscript.
Non-scientific names such as mosquitoes should not be italicised (line 36).
Different versions of English language/spelling were used, for example, parasitemia, favor (American English) and parasitaemia (British English). The authors should use one style of the English language.
The tables were presented in different styles, with and without borders. Stick to one style or the journal’s recommended style.
It will be very good if the authors can reduce the level of similarity from 25% to less than 20%.
Introduction
The prevalence of malaria in the two study sites should be stated.
Materials and Methods
Line 91: Add the location or country of manufacture.
In Section 2.5, minutes were spelt in full while in Section 2.6, it was abbreviated. Stick to one style to ensure uniformity.
Two different styles/symbols were used for the microlitre. Use one style/symbol.
For repeatability and reproducibility purposes, the authors should indicate the quality (concentration) of the genomic DNA used.
The type (and concentration) of the buffer used for making the gel and running the electrophoresis should also be indicated.
Other comments and corrections have been indicated in the reviewed manuscript.

There is no problem with the quality of the English language. Just minor editing is needed.
Author Response
General comment: The study is well written although there are a few grammatical, typographical and punctuation errors and these have been indicated in the reviewed manuscript.
Non-scientific names such as mosquitoes should not be italicised (line 36).
Different versions of English language/spelling were used, for example, parasitemia, favor (American English) and parasitaemia (British English). The authors should use one style of the English language.
The tables were presented in different styles, with and without borders. Stick to one style or the journal’s recommended style. It will be very good if the authors can reduce the level of similarity from 25% to less than 20%.
It will be very good if the authors can reduce the level of similarity from 25% to less than 20%.
Response:
Thank you very much for your thorough review and valuable comments. We appreciate your attention to detail and your suggestions for improving the manuscript. We have made the following revisions based on your feedback:
- Grammatical, Typographical, and Punctuation Errors: We have carefully reviewed and corrected the grammatical, typographical, and punctuation errors as indicated in the reviewed manuscript.
- Non-Scientific Names: We have corrected the formatting for non-scientific names such as "mosquitoes," ensuring they are not italicized (line 36).
“Malaria remains a potentially fatal disease caused by parasites transmitted through the bite of Anopheles mosquitoes”
- Consistency in English Language: We have standardized the use of English language throughout the manuscript, choosing American English for consistency. Terms like " parasitaemia " and "favor" are now uniformly used.
- Table Formatting: We have revised the tables to adhere to a consistent style with borders, following the journal’s recommended style.
- Similarity Reduction: We have reviewed the content and made necessary adjustments to reduce the similarity index from 25% to less than 20%.
These changes have been incorporated in the revised manuscript. We believe these revisions have improved the clarity and quality of our manuscript. Thank you again for your insightful comments and suggestions.
Comment 1: Introduction: The prevalence of malaria in the two study sites should be stated.
Response 1: Thank you for your valuable comment. We have added the number of malaria cases in the two study sites to the Introduction section. This information provides important context for the study and enhances the reader's understanding of the significance of the research. The changes can be found on page 2, lines 82-83. Your suggestion has improved the comprehensiveness of our manuscript.
“In 2023, the number of malaria cases in Vietnam was reported to be 448, with Binh Phuoc accounting for 5 cases and Dak Nong for 4 cases”.
Comment 2: Materials and Methods: Line 91: Add the location or country of manufacture.
Response 2: Thank you for your valuable comment. We have added the location or country of manufacture for the specified items in the Materials and Methods section. This information has been included to provide clarity and completeness. The changes can be found on page 3, line 108.
“Each DHA–PPQ tablet contains 40 mg of DHA and 320 mg of PPQ and is commercially known as Arterakin®, produced by Central Pharmaceutical Factory No. 1 (Hanoi, Vietnam); batch number 17001”.
Comment 3: In Section 2.5, minutes were spelt in full while in Section 2.6, it was abbreviated. Stick to one style to ensure uniformity.
Response 3: Thank you for your attention to detail. We have ensured uniformity by using the abbreviated spelling "min" consistently throughout Sections 2.5 and 2.6. This change helps maintain consistency and clarity in the manuscript. The revisions have been made in the respective sections.
Comment 4: Two different styles/symbols were used for the microlitre. Use one style/symbol
Response 4: Thank you for your attention to detail. We have standardized the use of the symbol for microlitre (µL) throughout the manuscript to ensure consistency. This change helps maintain uniformity and clarity in the presentation of measurements. The revisions have been made in all relevant sections.
Comment: For repeatability and reproducibility purposes, the authors should indicate the quality (concentration) of the genomic DNA used. The type (and concentration) of the buffer used for making the gel and running the electrophoresis should also be indicated.
Response: Thank you for your valuable comment. We have indicated the quality (concentration) of the genomic DNA used in section 2.4, on page 4, lines 152-155.
“The concentration and quality of the extracted DNA were assessed using a NanoDrop 2000 spectrophotometer (Thermo Fisher, USA). DNA purity was evaluated by measuring the A260/A280 ratio, with a range of 1.7-2.0 generally considered pure for DNA.
Additionally, we have specified the type and concentration of the buffer used for making the gel and running the electrophoresis. These details provide important information for repeatability and reproducibility. The changes can be found on page 4, lines 176-177, and page 5, lines 191-192. Your suggestion has greatly improved the thoroughness and clarity of our methodology section.
“Electrophoresis was performed using 0.5X TBE buffer to ensure optimal separation and resolution of the DNA fragments”.
Thank you for your thorough review and valuable comments. We have carefully addressed all other comments and corrections indicated in the reviewed manuscript. These revisions have been incorporated to improve the accuracy, clarity, and overall quality of the manuscript. We appreciate your attention to detail and your contributions to enhancing our work.
Reviewer 3 Report
Comments and Suggestions for Authors
The study evaluates the efficacy of the DHA-Piperaquine regimen in treating uncomplicated falciparum malaria across Binh Phuoc and Dak Nong provinces in Vietnam. Results indicate a significant decline in the regimen's effectiveness, with increased treatment failure rates and the presence of drug resistance markers such as pfK13 mutations and pfpm2 gene amplifications. This decline suggests the need for revised national malaria treatment guidelines and highlights the importance of molecular monitoring in addressing drug resistance.
Line 213: Change "Table 3" to "Table 2."
In Table 2, the information suggests a misleading association between gender and malaria positivity. It would be more appropriate to place the gender details in the methods section instead of this table.
For Table 3, Day 2, the parasitemia percentage for Dak Nong should be "8" rather than "08." The same applies to some numbers in Table 5. Ensure a consistent format for numbers across all tables.
Line 227: The table title should include "after treatment" to clarify the experimental methods.
Lines 249-250: How was the recrudescence rate calculated? From which figure or table? The author should at least show the sample size for each rate.
Lines 259-267: These texts should be related to Table 5. Please specify this in the manuscript.
For Result 3.3, please specify Table 6 in some parts of the text in this section.
Author Response
Comment 1: Line 213: Change "Table 3" to "Table 2."
Response 1: Thank you for your careful review and for pointing out this error. We have corrected the reference from "Table 3" to "Table 2" on page 5, line 225. Your attention to detail has helped us improve the accuracy of our manuscript. This correction has been made in the revised manuscript.
“The general characteristics of the study population are presented in Table 2”.
Comment 2: In Table 2 the information suggests a misleading association between gender and malaria positivity. It would be more appropriate to place the gender details in the methods section instead of this table.
Response 2: We highly appreciate this comment. During the preparation of the manuscript, we decided to include the gender information in Table 2 because it follows the protocol requirement outlined by the WHO guidelines for demographic characteristics. Our aim in managing Table 2 is to describe the demographic characteristics of the study population, including gender, which is a standard practice in reporting clinical study demographics. By including the gender details in Table 2, a comprehensive overview of the study population as per the WHO protocol can be realized. This allows for a clearer understanding of the demographic distribution of the participants, ensuring that all necessary demographic data are transparently presented. Additionally, this inclusion helps maintain consistency with the standardized reporting practices recommended by WHO guidelines. We believe that this approach does not imply any misleading association between gender and malaria positivity, but rather provides necessary context for understanding the study population.
Comment 3: For Table 3 Day 2 the parasitemia percentage for Dak Nong should be "8" rather than "08." The same applies to some numbers in Table 5. Ensure a consistent format for numbers across all tables.
Response 3: Thank you for your valuable comment. We appreciate your attention to detail and for pointing out these formatting inconsistencies. We have corrected the parasitemia percentage for Dak Nong from "08" to "8" in Table 3 and have ensured a consistent format for numbers across all tables, including Table 5. Your input has helped us improve the clarity and accuracy of our manuscript. These corrections can be found on page 6, Table 3 and page 8, Table 5 in the revised manuscript.
Comment 4: Line 227: The table title should include "after treatment" to clarify the experimental methods.
Response 4: Thank you for your insightful comment. We have updated the table title to include "after treatment" to clarify the experimental methods. Your suggestion has improved the clarity and precision of our manuscript. The updated title is reflected on page 6, line 239 in the revised manuscript.
“Table 1: Temperature and parasitemia of all patients from day 1 to day 3 after treatment”
Comment 5: Lines 249-250: How was the recrudescence rate calculated? From which figure or table? The author should at least show the sample size for each rate.
Response 5: Thank you for your valuable comment. I have included the sample size for the recrudescence rate in the revised manuscript. The recrudescence rate can be calculated based on Figure 1. Figure 1 uses the Kaplan-Meier method to calculate the cumulative recrudescence rate. This method estimates the probability of recrudescence over the 42-day follow-up period, taking into account the exact timing of recurrence events and including censored data (patients lost to follow-up or withdrawn). The Kaplan-Meier method offers a more detailed estimate over time and handles censored data, reducing potential biases.
Thanks to your insightful comment, I identified a calculation error due to data typing mistakes. This has been corrected in the revised manuscript. These details have been clarified in the revised manuscript on page 7, lines 262-263.
“Specifically, the data indicate that the recrudescence rate in Binh Phuoc reached 23.7% (9 out of 38), while in Dak Nong, it was slightly lower at 20% (3 out of 15).
Comment 6: Lines 259-267: These texts should be related to Table 5. Please specify this in the manuscript.
Response 6: We have specified that the texts in lines 259-267 are related to Table 5. This clarification can be found on page 8, lines 269-270.
“The treatment outcomes at the two study sites are presented in Table 5’”.
Comment 7: For Result 3.3 please specify Table 6 in some parts of the text in this section.
Response 7: We have specified Table 6 in the relevant parts of the text in Result 3.3. These changes can be found on page 9, Result 3.3 section, line 287-288.
“ Table 6 shows the prevalence of the pfK13 gene mutation and the copy number variation of the pfpm2 gene at the two study sites”.